# Mineralocorticoid Receptor May Regulate Glucose Homeostasis through the Induction of Interleukin-6 and Glucagon-Like peptide-1 in Pancreatic Islets

**DOI:** 10.3390/jcm8050674

**Published:** 2019-05-14

**Authors:** Rieko Goto, Tatsuya Kondo, Kaoru Ono, Sayaka Kitano, Nobukazu Miyakawa, Takuro Watanabe, Masaji Sakaguchi, Miki Sato, Motoyuki Igata, Junji Kawashima, Hiroyuki Motoshima, Takeshi Matsumura, Seiya Shimoda, Eiichi Araki

**Affiliations:** 1Department of Metabolic Medicine, Faculty of Life Sciences, Kumamoto University, Kumamoto 860-8556, Japan; t-kondo@gpo.kumamoto-u.ac.jp (T.K.); zenmaiing@yahoo.co.jp (K.O.); ramuramusayaka@yahoo.co.jp (S.K.); nomiyakawa@yahoo.co.jp (N.M.); w_takuro_0420@yahoo.co.jp (T.W.); masajisakaguchi@gmail.com (M.S.); satoum@kuh.kumamoto-u.ac.jp (M.S.); iga@gpo.kumamoto-u.ac.jp (M.I.); junjikawa@mac.com (J.K.) hmoto@gpo.kumamoto-u.ac.jp (H.M.); takeshim@gpo.kumamoto-u.ac.jp (T.M.); 2Food and Health Sciences, Prefectural University of Kumamoto, Kumamoto, 862-8502, Japan; sshimoda@pu-kumamoto.ac.jp

**Keywords:** Mineralocorticoid receptor (MR), Glucagon-like peptide-1 (GLP-1), interleukin-6 (IL-6), α-cells, glucose homeostasis

## Abstract

Because the renin-angiotensin-aldosterone system influences glucose homeostasis, the mineralocorticoid receptor (MR) signal in pancreatic islets may regulate insulin response upon glucose load. Glucagon-like peptide-1 (GLP-1) production is stimulated by interleukin-6 (IL-6) in pancreatic α-cells. To determine how glucose homeostasis is regulated by interactions of MR, IL-6 and GLP-1 in islets, we performed glucose tolerance and histological analysis of islets in primary aldosteronism (PA) model rodents and conducted in vitro experiments using α-cell lines. We measured active GLP-1 concentration in primary aldosteronism (PA) patients before and after the administration of MR antagonist eplerenone. In PA model rodents, aldosterone decreased insulin-secretion and the islet/pancreas area ratio and eplerenone added on aldosterone (E+A) restored those with induction of IL-6 in α-cells. In α-cells treated with E+A, IL-6 and GLP-1 concentrations were increased, and anti-apoptotic signals were enhanced. The E+A-treatment also significantly increased MR and IL-6 mRNA and these upregulations were blunted by MR silencing using small interfering RNA (siRNA). Transcriptional activation of the IL-6 gene promoter by E+A-treatment required an intact MR binding element in the promoter. Active GLP-1 concentration was significantly increased in PA patients after eplerenone treatment. MR signal in α-cells may stimulate IL-6 production and increase GLP-1 secretion, thus protecting pancreatic β-cells and improving glucose homeostasis.

## 1. Introduction

Several large-scale clinical trials have shown that angiotensin II receptor antagonist has significant preventative effects for the onset of diabetes [1]. Results from these trials and other studies have thus demonstrated that the renin angiotensin aldosterone system has an impact on glucose tolerance through modification of insulin resistance and/or insulin secretion [2]. In humans, primary aldosteronism (PA), a leading cause of secondary hypertension, has been associated with diabetes due to impaired insulin sensitivity and/or insulin secretion [3]. Previous research has suggested that aldosterone decreases glucose-stimulated insulin secretion in vivo and in vitro (in murine islets) [4]. In addition, sub-chronic stimulation of the mineralocorticoid receptor (MR), the receptor for aldosterone, protects pancreatic β-cells against glucocorticoid-induced lower cytosolic Ca^2+^ responses to glucose [5]. 

A previous report proposed that the administration of interleukin-6 (IL-6) or elevated IL-6 concentrations in response to exercise stimulates incretin hormone glucagon-like peptide 1 (GLP-1) secretion from intestinal L cells and pancreatic α-cells in humans [6]. GLP-1 has been shown to lead to glucose-dependent insulin secretion, induction of β-cell proliferation and enhanced resistance to β-cell apoptosis [7]. Further, a potential role for local IL-6 in the regulation of α-cell growth and function during neonatal development has been demonstrated [8], indicating that local IL-6 may exert beneficial effects on the functional integrity of islets. Indeed, IL-6, IL-6 receptor and glucagon are co-expressed in pancreatic α-cell populations during development in murine islets [8]. Although MR signals may regulate pancreatic islet function, the role of MR in the induction of IL-6 and GLP-1 in islet cells has not been elucidated. The purpose of this study was to clarify the effects of MR signal on pancreatic islets, especially in association with IL-6 and GLP-1. 

## 2. Experimental Section

### 2.1. Materials Establishment of PA Model Rodents, Treatment Groups and Histological Analysis of Pancreatic Islets

Male Sprague-Dawley rats (*n* = 18), obtained from CLEA Japan Inc. (CLEA Japan Inc., Tokyo, Japan), were housed in the Center for Animal Resources and Development of Kumamoto University. The experimental procedures were approved by the Animal Experimentation Ethics Committee of Kumamoto University (B24-129). 

The animals were placed in one of the following 28-day treatment protocols: (1) control (vehicle/normal chow) (*n* = 6); (2) aldosterone infusion (aldosterone/normal chow) (*n* = 6); or (3) aldosterone infusion + oral eplerenone treatment (aldosterone/eplerenone chow) (*n* = 6). We used an Alzet 2004 osmotic mini-pump (Alza Corp., Palo Alto, CA, USA) to administer either vehicle (9% ethanol/87% propylene glycol/4% dH_2_O) or 2.9 mg/mL d-aldosterone (Sigma Chemical Company, St. Louis, MO, USA), as previously reported [9]. Eplerenone (provided from Pfizer Inc., New York City, NY, USA) was administered in the chow obtained from Kyudou Ltd. (Saga, Japan) at a concentration of 1.2 mg eplerenone/g of chow, resulting in an approximate dose of 100 mg/kg/day, according to the previous report [9]. During the experiments, all rats drank 1% saline (wt/vol.) for 4 weeks prior to the study. 

On day 28, plasma aldosterone concentration (PAC), blood pressure, body weight and serum potassium level of each treatment group were measured. Glucose tolerance was examined in each treatment group using the 2 g/kg intraperitoneal glucose tolerance test (ipGTT) on day 28.

After the treatments, animals were anesthetized with 50 mg/kg pentobarbital and 30% isoflurane diluted with propylene glycol (Wako, Japan) and exsanguinated. The pancreas from each rat was isolated and routinely processed and embedded in paraffin or immediately snap-frozen in liquid nitrogen. Pancreas sections (3 μm) from paraffin blocks were dyed with Masson’s trichrome staining. Other pancreas sections (5 μm) from frozen blocks were immunostained following standard procedures using primary antibodies for IL-6 (No. 18611, Immuno-Biological Laboratories Co., Ltd., Gunma, Japan), glucagon (sc-13091, SANTA CRUZ BIOTECHNOLOGY, Dallas, TX, USA), MCP-1 (No. 18371, Immuno-Biological Laboratories Co.), Iba1 (Wako, Japan) and CD206 (sc-9139, SANTA CRUZ BIOTECHNOLOGY). The size of each pancreatic islet area was measured and compared using BZ-9000 (Keyence Corp., BIOREVO, Chicago, IL, USA). 

### 2.2. Cell Lines and Treatments

The 1C3 IKEI pancreatic cell line was kindly provided by H. Ishikawa (Nippon Dental University entrusted to RIKEN BRC, Japan). Alpha TC1 clone 6 cells (αTC cells) (ATCC, Manassas, VA, USA) and the 1C3 IKEI cells were cultured in Dulbecco’s Modified Eagle Medium (DMEM) containing 25 mmol/L glucose with 10% steroid-free FBS (Sigma Aldrich, St. Louis, MO, USA) and antibiotics (100 μg/mL streptomycin, 100 units/mL penicillin, 0.25 μg/mL amphotericin B), in the presence or absence of 10^−7^ M aldosterone, 10^−5^ M eplerenone or a combination of the two treatments [10]. Cells were cultured at 37 °C in a humidified atmosphere with 5% CO_2_. 

### 2.3. Glucose Homeostasis of PA Patients before and after Treatment 

This study is a prospective study and enrolled 13 patients who were diagnosed with PA at the Kumamoto University hospital between 2014 and 2017. In accordance with the diagnosis of PA described by the Japan Endocrine Society, the diagnosis was performed using the result of plasma aldosterone concentration (pg/mL) / plasma renin activity (ng/mL/hr) ratio (ARR > 200), the captopril suppression test (ARR > 200 at 60 min after loading 50mg of captopril) and the adrenal vein sampling. Glucose tolerance was examined in the 13 patients using the 75 g oral glucose tolerance test (75 g OGTT) before and after the treatments of PA (one to three months after eplerenone administration (50 mg average per day)). We used the 2006 WHO recommendations for the diagnostic criteria for normal glucose tolerance (fasting plasma glucose (FPG) < 110 mg/dL and 2 hour plasma glucose (2-h PG) < 140 mg/dL), impaired glucose tolerance (FPG 110–125 mg/dL and 2-h PG 140–199 mg/dL) and diabetes mellitus (FPG > 126 mg/dL or 2-h PG > 200 mg/dL). Active GLP-1 concentration was determined 60 min after the glucose load. 

The study protocol was confirmed by the ethical guidelines of the Declaration of Helsinki and written informed consent was obtained from each patient. This research was approved by the Ethics Review Committee at Kumamoto University (Advanced Ethics No. 1956). Clinical trials were registered with an approved International Committee of Medical Journal Editors (ICMJE) clinical trial registry, University Hospital Medical Information Network (UMIN) clinical trials registry (ID: UMIN000017666). 

### 2.4. ELISA Assays 

Insulin, IL-6 or active GLP-1 concentrations were assayed with commercial ELISA kits according to the manufacturer’s instructions (LBIS Insulin-Rat-T, FUJUFILM Wako Pure Chemical Corporation, Osaka, Japan; hIL-6 QKit, R&DSYSTEMS, Minneapolis, MN, USA; IL-6 ELISA kit, Mouse, Proteintech Group, Inc., Rosemont, IL, USA; GLP-1, Active Form Assay Kit, Immuno-Biological Laboratories Co., Ltd., Fujioka-Shi, Gunma, Japan). 

### 2.5. RT-PCR

cDNA was synthesized from total RNA prepared from cells using the RNeasy Mini Kit (QIAGEN, Hilden, Germany). Expression levels of genes were determined by reverse transcription PCR (RT-PCR) using the ReverTra Ace® qPCR RT Master Mix (TOYOBO, Kita-ku, Osaka, Japan) aler Fast-Start DNA Master Plus SYBER Green (Roche Diagnostics, Basel, Switzerland) on a LightCycler 2.0 (Roche Diagnostics). Glyceraldehyde-3-phosphate dehydrogenase (GAPDH) mRNA was used as a normalization control for RNA quality, RNA quantity and RT method. Primer sequences are shown in Appendix A. 

### 2.6. siRNA Transfection 

On the day before transfection, αTC1 cells were seeded in 6-well plates and cultured overnight until cells reached 75% confluence. Cells were transiently transfected with siRNA (100 pmol/well, Stealth siRNA, Santa Cruz) using Lipofectamine 2000 (5 μL/well, Invitrogen, Paris, France) mixed with OptiMEM serum-free medium (Invitrogen, Paris, France). Four hours after transfection, cells were washed twice with phosphate buffered saline (PBS) and then cultured in the presence or absence of a combination of the aldosterone and eplerenone treatments. The siRNA sense sequence (siMR, siGR) and the control sense sequence (CONMR, CONGR) for MR and GR are shown in Appendix A. 

### 2.7. Reporter Plasmids Construction and Luciferase Assay 

The human MR expression vector (pRR-MR-5Z) and the human GR expression vector (pk7-GR-GFP) were obtained from Charles Miller (No. 23059, No. 15534, Addgene, Cambridge, MA, USA) [11,12]. 

The IL-6 promoter region was amplified by PCR from human genomic DNA (QuickGene, Kurabo) for the construction of the IL-6 Luc plasmid. The IL-6 Mutant Luc plasmid, containing mutations in the MR binding element (MRE), was constructed by inserting pairs of complementary hybridized oligonucleotides. The list of qPCR primers for IL-6 promoter and sense sequence for mutated MRE are shown in Appendix A. The IL-6 promoter sequences were then ligated into the PGL-3 vector after digestion with KpnI and XhoI to generate luciferase reporter vectors. Luciferase assay was performed using the Dual-Luciferase® Reporter Assay System (Promega) according to the manufacturer’s instructions. 

### 2.8. Statistical Analysis 

Statistical analysis was performed with SPSS software (IBM, Chicago, IL, USA). All values were expressed as mean ± standard deviation (S.D.). Data were evaluated using Chi-square tests or paired t-tests if data were normally distributed or a Wilcoxon signed-rank test if not. Sequential changes were analyzed by repeated-measures ANOVA. Two-sided *p*-values of less than 0.05 were considered to indicate statistical significance. 

## 3. Results 

### 3.1. Eplerenone Treatment Protects Islet Cells from Cell Damage Caused by Aldosterone 

To examine the effects of MR signals in pancreatic islets, we generated a rodent PA model by aldosterone infusion and examined the impact of eplerenone treatment. Sprague Dawley rats were randomly placed in one of the following 28-day treatment groups: vehicle (Control), aldosterone infusion (2.9 mg/mL) (Aldosterone) or aldosterone infusion (2.9 mg/mL) with eplerenone administration (100 mg/day/kg) (E+A). 

Plasma aldosterone concentration was significantly increased in the Aldosterone group compared with the Control group or E+A group (Figure 1A). Systolic and diastolic blood pressure values were significantly increased in the Aldosterone group compared with the Control group or E+A group (Figure 1B). There was no significant difference in body weight (Figure 1C). Serum potassium level was significantly lower in the Aldosterone group than that in the Control group or the E+A group (Figure 1D). 

Blood glucose was higher in the Aldosterone group than that in the Control group or the E+A group (Figure 1E). Area under the curve (AUC) of blood glucose concentration during ipGTT was significantly increased in the Aldosterone group compared with the Control group or the E+A group (Figure 1F). IRI was lower in the Aldosterone group than that in the Control group or the E+A group (Figure 1G). AUC of IRI during ipGTT was significantly decreased in the Aldosterone group compared with the Control group (Figure 1H).

Histological analysis revealed that the islet/pancreas area ratio was significantly reduced in the Aldosterone group compared with the Control group and was completely restored in the E+A group (Figure 2A–C,G). 

The percentage of the α-cell area was significantly increased in the Aldosterone group compared with the Control group or E+A group (% α-cell area in islet area: 24.6% ± 5.9% vs. 45.5% ± 5.1% vs. 17.3% ± 3.0% in Control, Aldosterone and E+A groups, respectively; *p* < 0.05) (Figure 2D–F,H). 

M1 macrophages were accumulated in pancreatic β-cells of Aldosterone group (Appendix A). 

To confirm the inflammatory cytokines, which accelerate the accumulation of M1 macrophages, MCP-1 and IL-6 were immunostained in pancreatic islets. Interestingly, although MCP-1 was expressed in the pancreatic β cells of the Aldosterone group (Appendix A), IL-6 was expressed in the α-cells. In addition, although a small number of IL-6-positive pancreatic α-cells were observed in Control islets, IL-6-positive cells were diminished in islets of the Aldosterone group. Upon E+A treatment, the percentage of IL-6-positive α-cells were significantly increased compared with those in the Control or Aldosterone groups (% IL-6-positive α-cells in α-cells: 45.6% ± 10.2% vs. 7.5% ± 6.3% vs. 80.7% ± 5.8% in Control, Aldosterone and E+A groups, respectively; *p* < 0.05) (Figure 2D–F,I). These data indicate that IL-6 expression in pancreatic α-cells was reduced by the treatment with aldosterone and the treatment of eplerenone restored the IL-6 expression. 

In addition, X-box binding protein 1s (XBP1-s) was significantly increased in the pancreas of the Aldosterone group compared with that of the Control or E+A groups. As XBP1-s is a key modulator of unfolded protein response (UPR), these data suggest that endoplasmic reticulum (ER) stress in the pancreas was increased in PA model rodents (Appendix A). 

### 3.2. Eplerenone Treatment Increases MR, IL-6 and Active GLP-1 Secretion from α-Cells 

A previous report suggests that IL-6 and glucagon are co-expressed in pancreatic α-cell populations during development in murine islets and IL-6 exhibited developmental regulation in the rat pancreas [8]. IL-6 expressed in α-cell may regulate islet/pancreas area ratio, thus we investigate IL-6 and GLP-1 effects and MR/IL-6 axis in α-cells. In addition, previous studies have suggested that IL-6 stimulates active GLP-1 secretion, which is liberated from its precursor proglucagon through processing by the enzyme prohormone convertase (PC) 1/3 [6]. To investigate whether production of IL-6 and active GLP-1 is increased by eplerenone in α-cells, αTC cells were stimulated with vehicle (Control), 10^−7^ M aldosterone (Aldosterone) or 10^−7^ M aldosterone after pre-treatment with 10^−5^ M eplerenone for 1 h (E+A). Serum and glucocorticoid-regulated kinase 1 (SGK1) mRNA which is activated by aldosterone, was significantly increased in the Aldosterone group and partially restored by E+A (Appendix A). The MR mRNA level was increased by 1.6-fold in the E+A treatment group compared with the Control group but no changes were observed when the cells were treated with aldosterone alone (Figure 3A). IL-6 mRNA was also increased by 3-fold upon E+A treatment compared with the control and PC1/3 mRNA showed a similar induction pattern to IL-6 (Figure 3B,C). 

To investigate the effects of eplerenone alone, αTC cells (p8) were stimulated with vehicle (Control), 10^−5^ M eplerenone alone (Eplerenone) or 10^−7^ M aldosterone after pre-treatment with 10^−5^ M eplerenone for 1 h (E+A). IL-6 mRNA was increased by 2.2-fold upon eplerenone treatment compared with the control and PC1/3 mRNA showed a similar induction pattern as IL-6 (Appendix A). 

Positive correlations between MR mRNA and IL-6 mRNA as well as PC1/3 mRNA and IL-6 mRNA levels were observed (Figure 3D). Furthermore, while both IL-6 and active GLP-1 concentrations were significantly decreased in the supernatant of the Aldosterone group compared with the Control group (*p* = 0.03), both were significantly increased in the supernatant of the E+A group compared with the Aldosterone group (*p* = 0.04) (Figure 3E,F). We also examined the levels of several apoptosis-related genes. The mRNA level of the pro-apoptotic gene caspase-3 was significantly suppressed and the mRNA level of the apoptosis suppressor gene Bcl-2 was significantly increased in the E+A group compared with both the Control or Aldosterone groups (Figure 3G,H).

As a high aldosterone concentration (10^−7^ M aldosterone (Aldosterone)) may have some effects on the glucocorticoid receptor (GR), 10^−7^ M aldosterone may reduce IL-6 expression due to the GR signal. The experiment was conducted with 10^−8^ M aldosterone to confirm an adequate concentration of aldosterone. There is no difference in IL-6 mRNA expression level between 10^−7^ M aldosterone and 10^−8^ M aldosterone (Appendix A). 

### 3.3. MR regulates IL-6 expression in α-cells 

To investigate the relationship between MR and IL-6 in islet cells, MR expression was modulated in αTC cells using siRNA. When αTC cells were transfected with siRNA targeting MR (siMR), MR mRNA expression was decreased by 34% compared with the control siRNA (CONMR) (Figure 4A). While the control αTC cells showed an increased MR mRNA level by 1.3 fold upon E+A treatment (CONMR E+A) compared with untreated cells, MR mRNA level of the αTC cells transfected with siMR was significantly decreased, by 42%, upon E+A treatment compared with the CONMR E+A group. Furthermore, while IL-6 mRNA level was increased by 3.1 fold in the E+A treated cells transfected with CONMR, this induction of IL-6 mRNA was also blunted by knockdown of MR by siMR (Figure 4B). 

We also generated a human MR expression vector (MR+) and confirmed significantly increased MR mRNA expression, by 440-fold, upon transfection with the MR+ vector in another α-cell line, IKEI cells (Figure 4C), because IL-6 mRNA expression was not increased in αTC cells transfected with the MR+ vector (Appendix A). IL-6 mRNA expression was also significantly increased by 8.9-fold in IKEI cells transfected with the MR+ vector compared with controls (Figure 4D). 

MR is presumed to differentiate from GR in the process of evolution and the amino acid sequence of its DNA binding domain has 94% homology. Therefore, the DNA binding element of MR overlaps with that of GR. To investigate the relationship between GR and IL-6, we also modulated GR expression in αTC cells using siRNA and human GR expression vector (GR+) (Appendix A). GR (siGR) decreased GR mRNA expression level by 64% compared with the control siRNA (CONGR) (Appendix A). There was no difference in MR mRNA level and IL-6 mRNA level between siGR Control group and siGR E+A group (Appendix A). GR mRNA expression was increased by 2723-fold upon transfection with the GR+ vector in α-cells (Appendix A). Overexpression of GR in α-cells significantly decreased IL-6 mRNA expression (Appendix A).

### 3.4. Induction of IL-6 Expression by Eplerenone Requires MR Binding Element in the IL-6 Promoter 

To determine the mechanism of IL-6 mRNA induction by eplerenone treatment, luciferase assays were performed. A previous report showed that the IL-6 promoter contains a GR binding element (GRE) from −173 to −151 nucleotides upstream of the transcription initiation site [13]. As the amino acid sequence of DNA binding domain of MR and GR has 94% homology, GRE may also be act as MRE. We generated luciferase reporter vectors driven by either the wild-type IL-6 promoter containing MRE (IL-6 Luc) or by the mutant promoter in which the MRE was mutated (IL-6 Mutant Luc) (Figure 5A). When αTC cells were transfected with the wild-type IL-6 Luc, luciferase activity was significantly increased by E+A treatment compared with that of either the Control group or the Aldosterone group (Figure 5B). However, when αTC cells were transfected with IL-6 Mutant Luc, E+A failed to induce luciferase activity (Figure 5B). 

### 3.5. Oral Eplerenone Treatment Improves the Impairment of Glucose Homeostasis in Patients with PA

The experiments of PA model rodents and α-cell derived cells suggested that eplerenone administration enhanced active GLP-1 secretion from pancreatic α-cells and improved glucose metabolism. To confirm the active GLP-1 secretion in humans, we measured active GLP-1 concentration in PA patients before and after the eplerenone treatment. The baseline characteristics of the 13 patients with PA included in this study are shown in Figure 6. Glucose tolerance was determined in all cases. The ratio of patients with low potassium (serum potassium level less than 3.5 mEq/L, Merck Manual) was 23%. The frequency of patients with impaired glucose tolerance (IGT) was 77% (Figure 7A). In 13 patients who were treated with the oral MR antagonist eplerenone, active GLP-1 concentration was determined one to three months after the treatment.

Homeostasis model assessment for insulin resistance (HOMA-R) was significantly increased after the treatments (*n* = 13; 1.34 ± 0.89 vs. 1.88 ± 0.89, *p* = 0.04). There were no differences in the insulinogenic index, HbA1c, ΣBG and ΣIRI before and after the treatments (*n* = 13; 0.63 ± 0.43 vs. 0.59 ± 0.38; *p* = 0.70, 5.43 ± 0.2 vs. 5.4 ± 0.3; *p* = 0.60, 739.6 ± 86.1 vs. 760.9 ± 95.1; *p* = 0.46, 266.3 ± 212.8 vs. 272.3 ± 191.5; *p* = 0.78). The treatments showed a trend of improved blood glucose concentration at 120 min during 75 g OGTT (148.8 ± 21.1 vs. 142.9 ± 19.1 mg/dL; *p* = 0.23) and resulted in a significant increase of normal glucose tolerance (NGT) response in PA patients (*n* = 13, from 23% to 38%) (*p* = 0.0004, Chi-square test) (Figure 7A,B). Insulin concentration at 60 min during 75 g OGTT showed a trend of increase after the treatments (*n* = 13, 62.7 ± 53.1 vs. 69.3 ± 46.7 μU/mL; *p* = 0.30) (Figure 7C). Interestingly, active GLP-1 concentration at 60 min during OGTT was significantly increased after eplerenone treatment (*n* = 13, 2.74 ± 1.32 vs. 4.56 ± 2.39 pmol/L; *p* < 0.05) (Figure 7D). There were no correlations between GLP-1 and HOMA-R when compared per patient (Figure 7E). 

Together, these results indicate that the treatments of PA with eplerenone may increase active GLP-1 secretion, which in turn contributes to improving glucose homeostasis in PA patients. 

## 4. Discussion

In this study, we investigated the role of MR on GLP-1 secretion and glucose homeostasis in islets through an in vivo animal model and in vitro cell line experiments and a human clinical study. In the PA model rodents, IL-6 expression was induced in α-cells of pancreatic islets upon eplerenone treatment. This eplerenone-dependent IL-6 induction requires specific binding elements for MR (MREs) in the IL-6 gene promoter. We found that the oral eplerenone treatment increased active GLP-1 secretion and reduced the number of patients with IGT. The MR-dependent IL-6 induction contributes to increase active GLP-1 levels through the induction of PC1/3 and thus may protect islets from metabolic insults (Figure 8).

To investigate how glucose homeostasis is regulated by MR in pancreatic islets cells, we examined ipGTT in the PA model rodents. We found that glucose AUC was increased and IRI AUC was decreased during ipGTT in the Aldosterone group of rodents (Figure 1). As the result suggested that the eplerenone treatment somehow improved insulin secretion, we carried out histological analysis of pancreatic islets in the PA rodent model to further determine the detailed mechanism of increased insulin secretion by eplerenone. 

In histological examination, MCP-1 was expressed in pancreatic β-cells and M1 macrophages were accumulated in the Aldosterone group. Inflammation in β-cells by aldosterone may cause more substantial effects in PA model rodents. We also found that the pancreatic islet area ratio was reduced in the Aldosterone group and was restored by E+A treatment, in parallel with the induction of IL-6 in α-cells of pancreatic islets (Figure 2). Recent reports showed that IL-6 stimulates GLP-1 secretion from α-cells by the activation of PC1/3 to convert from precursor proglucagon into mature GLP-1 [6]. Indeed, IL-6 as well as active GLP-1 induction were observed by E+A in our experiments (Figure 3). 

In addition, we found that endoplasmic reticulum (ER) stress in the pancreas was increased in PA model rodents and was restored by E+A treatment. Recent reports have suggested the importance of α-cell-derived GLP-1 for glucose homeostasis during aging and metabolic stress [14]. ER stress in pancreatic islets was reported as a mediator of nitric oxide-induced apoptosis [15] and an age-dependent modifier of islet survival and function [16]. Because ER stress could be suppressed by GLP-1 [17], treatment with eplerenone in PA model rodents may protect pancreatic β-cells through reduced ER stress by increased secretion of active GLP-1 from α-cells (Figure 8, #6). 

To confirm the induction of IL-6 and increased secretion of GLP-1 by eplerenone treatment in α-cells, we performed experiments using α-cell-derived cell lines. First, we found that MR mRNA expression was significantly increased in αTC cells by E+A treatment. Although the mechanism of eplerenone-induced MR upregulation was not elucidated in this study, we speculate that an eplerenone-MR complex may bind to MRE in P2 promoter of the MR gene in α-cells and promote MR expression (Figure 8, #1). A previous study reported the presence of two functional promoters in the human MR gene—referred to as P1 and P2—in the 5′-flanking regions [18]. Although aldosterone was shown to stimulate both P1 and P2 activity in murine embryonic stem cell lines, MR knock down inhibits aldosterone-induced P2 promoter activity [19], indicating that P2 could contain a functional MRE. Whether an eplerenone-MR complex binds to the MRE of P2 in α-cells, remains to be elucidated. The functional association and characterization of MR and the MRE in the MR gene should be investigated in future studies. 

Second, we confirmed that MR induces IL-6 by binding with IL-6 promoter (Figure 8, #2). Our in vitro studies also revealed that the expression level of MR and IL-6 mRNA showed a similar pattern and correlated positively with each other. The discrepancy observed between increased IL-6 and GLP-1 gene expression and reduced release in supernatants may be due to the fact that the mechanisms regulating the proinflammatory cytokines, without signal peptide, are not fully understood. Previous reports revealed that microvesicle shedding is a major secretory pathway for rapid IL-1β release from activated monocytes and may represent a more general mechanism of the secretion of similar leaderless secretory proteins [20]. In our study, IL-10 mRNA expression was significantly increased in the Aldosterone group compared with the control and the E+A group in αTC cells (Appendix A). IL-10 altered the number of autophagosome (microvesicle) in myocytes treated with Angiotensin II [21]. In addition, previous reports suggest that the activation of IL-10 receptors leads to the inhibition of IL-6 release [22,23]. Thus IL-6 and GLP-1 release in the supernatants may be decreased in the Aldosterone group. More detailed mechanisms should be clarified in future. Notably, a previous report showed that the IL-6 promoter contains a GRE [13]. These data suggest that an eplerenone-MR complex may bind to the GRE (= MRE) in the IL-6 promoter. 

We further investigated the mechanism of MR regulation of IL-6 expression in α-cells. The eplerenone-stimulated induction of IL-6 appeared to be MR expression dependent as shown in our siRNA data. In addition, overexpression of MR alone increased the level of IL-6 mRNA, indicating that unbound MR may directly associate with MRE to activate IL-6 expression, because these experiments were performed in a glucocorticoid- and mineralocorticoid-free environment. Indeed, a previous report showed that MR overexpression in ES cell-derived cardiomyocytes increased beating frequency without aldosterone, suggesting that cardiac MR is at least partially activated in a ligand-independent manner [24]. Furthermore, we performed luciferase assays using IL-6 gene promoter with or without the potential MRE in α-cells. Interestingly, E+A but not aldosterone alone activated the IL-6 promoter, suggesting that the eplerenone-MR complex or MR alone may drive IL-6 expression through the MRE. Although the GR-binding site of the IL-6 promoter between -173 to -145 lack discernible inverted repeats, sequences show similarity with a negative GRE sequence [13]. GRs can remain as monomers and interact with transcription factors such as activating protein-1 (AP-1) or nuclear factor-κB [25]. Monomer MRs was increased when the α-cells were treated with eplerenone. The monomer MRs may be able to interact with MRE in the absence of ligand. However, IL-6 mRNA expression did not increase in αTC cells transfected with human MR-expressing vector (MR+). This suggests that human MR may not bind with the MRE of the IL-6 promoter in murine α-cells. In addition, GR did not mediate the increase of IL-6 mRNA expression in α-cells (Appendix A). 

Third, IL-6 may activate PC1/3 promoter (Figure 8, #3). We confirmed that IL-6 and PC1/3 mRNA levels correlated with each other in α-cells. IL-6 secretion and active GLP-1 secretion from α-cells are both significantly increased upon E+A treatment. Indeed, a previous report showed that IL-6 treatment increased the expression of GLP-1 and PC1/3 mRNA in human α-cells [6]. In addition, a recent study showed that IL-6 significantly increased both PC1 protein and mRNA in mouse corticotroph AtT-20 cells through a Janus kinase (JAK)-signal transducers and activator of transcription (STAT)-related pathway, which is involved in the transcriptional regulation of PC1 gene expression through a PC1 promoter regulatory element [26]. 

Fourth, PC1/3 may induce active GLP-1 secretion (Figure 8, #4). We verified that E+A treatment significantly increased GLP-1 secretion compared with aldosterone treatment in the α-cell line. A previous report showed that induced PC1/3 cleaves proglucagon into GLP-1 in α-cells, resulting in elevated plasma levels of GLP-1 [27]. 

Finally, active GLP-1 may protect islet cells from apoptosis (Figure 8, #5 and #6). To investigate the mechanism of cell protection in pancreatic islets, we measured apoptosis-related genes in α-cells. Caspase-3 mRNA significantly decreased, and Bcl-2 mRNA significantly increased, upon E+A treatment, indicating that treatment with eplerenone may protect α-cells from apoptosis. Previous reports have identified a preventative effect of eplerenone from apoptosis in glomerular cells and cardiomyocytes [28,29]. In addition, one report suggested that GLP-1 treatment increased the expression of Bcl-2 and reduced the levels of caspase-3 in retinal cells in type 2 diabetic rats through the GLP-1 receptor-extracellular signal regulated kinase 1/2-histone deacetylase 6 (GLP-1R-ERK 1/2-HDAC 6) signaling pathway [30]. Indeed, a recent report detected GLP-1R in α-cells and chronic exposure to GLP-1 increased GLP-1 synthesis from αTC cells [31]. Thus, eplerenone treatment may protect α-cells from apoptosis through autocrine signaling of GLP-1. 

We confirmed that pancreas/duodenum homeobox 1 (Pdx-1) mRNA expression was significantly decreased and caspase-3 mRNA expression was significantly increased in MIN6 cells treated with aldosterone compared with the control and E+A groups (Appendix A). We previously showed that the GLP-1 agonist Exendin-4 upregulated Pdx-1 in regenerating α-cells [32]. In addition, several studies have suggested that increased activation of GLP-1R in α-cells is beneficial for insulin secretion, proliferation and anti-apoptosis activities of the β-cell [17,33,34,35]. Together these findings indicate that eplerenone treatment increases GLP-1 secretion in α-cells and the GLP-1 could protect β-cells from apoptotic signals (Figure 8, #6). 

Considering that glucocorticoids bind not only the glucocorticoid receptor but also MR with comparable affinity to aldosterone, and glucocorticoids have a 102–103-fold higher blood concentration compared with aldosterone, glucocorticoids have to be converted to the deactivated form by 11β-hydroxysteroid dehydrogenase type 2 (11β-HSD2) to retain the specific action of aldosterone by MR [10]. Our data showed that 11β-HSD2 mRNA was significantly increased in cells treated with eplerenone (Appendix A), suggesting that islets are mainly regulated by MR signals but not glucocorticoid receptor signals. A recent report revealed that 11β-HSD1 mRNA is essentially undetectable in both α-cells and cells in purified human islets [36]. In addition, 11β-HSD2 mRNA was detectable at much lower levels in pancreatic islets and the MIN6 cell line than that in kidney [4]. However, as a high fat diet induces a modest elevation of 11β-HSD1 in the islets of rodents [37], the expression level of 11β-HSD may be regulated by nutritional environment of islets. Since only few reports have examined 11β-HSD2 in islet cells, more detailed mechanisms must be investigated in future studies. 

To explain the mechanism of impaired glucose homeostasis in human PA patients, impaired insulin secretion due to hypokalemia was initially described [38] and another study discussed insulin resistance in PA subjects [39]. Although 77% of PA subjects in this study showed impairment of glucose tolerance, the frequency of patients with hypokalemia was only 23%. This suggests that low potassium levels may only have a small influence on glucose intolerance in PA. In addition, as homeostasis model assessment insulin resistance (HOMA-R) was significantly increased after treatment of PA patients with oral eplerenone administration, insulin resistance may not be the major pathophysiology of glucose intolerance in PA patients. In a previous report, active GLP-1 at 60 min after glucose load in Japanese non-obese healthy subjects was around 8 pmoL/L [40]. In our data, the active GLP-1 secretion at 60 min after glucose load in PA patients (2.7 pmoL/L) was much lower, suggesting the impaired GLP-1 secretion in PA patients. Furthermore, active GLP-1 concentration was significantly increased after eplerenone treatment in this study. In addition, active GLP-1 concentration was significantly decreased in the supernatant of αTC cells which were treated with aldosterone, thus aldosterone/MR-mediated decrease of GLP-1 secretion may be one of the main contributors to the impairment of glucose homeostasis in PA patients. Recent studies have shown that GLP-1 concentration is lower in type 2 diabetic patients than that in healthy subjects [41,42]. Salt intake affects the serum GLP-1 level in normotensive salt-sensitive subjects [43]. In addition, MR blockade in healthy adults tended to improve postprandial glucose concentration and insulin secretion [44]. Indeed, normalization of GLP-1 secretion by MR blockade suppressed prolonged hyperglycemia. Normal GLP-1 secretion may protect islet cells in the long term. This study is the first to elucidate further details of the relationship between MR and glucose homeostasis by measuring active GLP-1 response to oral glucose load before and after oral eplerenone treatment in PA patients. 

## 5. Conclusions

In summary, we have identified a novel mechanism by which the MR signal exerts a protective function in rodent and human islets through the engagement of IL-6 induction, PC 1/3 activation and the active GLP-1 secretion pathway (Figure 8). This illustrates the toxicity of aldosterone excess in the pathogenesis of glucose intolerance in PA. Eplerenone may have a potential benefit for diabetic therapy with its effect of increasing GLP-1 secretion in α-cells. Understanding this signaling in more detail may provide therapeutic alternatives for the treatment of diabetes. 

## Figures and Tables

**Figure 1 jcm-08-00674-f001:**
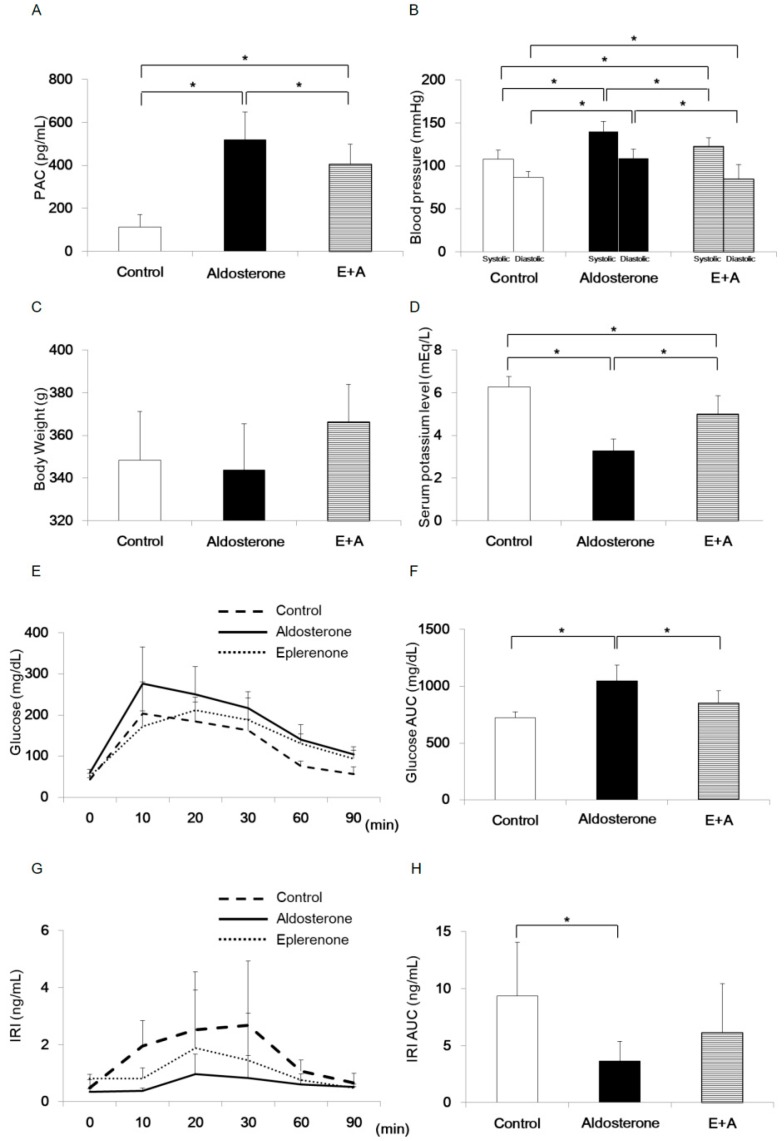
Comparisons of each parameter in primary aldesteronism (PA) model rodents on day 28. Rats were randomized into three treatment groups (*n* = 6 each group): control, treated with vehicle; aldosterone, treated with 2.9 mg/mL d-aldosterone; and E+A, treated with d-aldosterone and 100 mg/kg/day oral eplerenone administration. (**A**) Plasma aldosterone concentration of each treatment group. (**B**) Blood pressure value of each treatment group. (**C**) Body weight value of each treatment group. (**D**) Serum potassium level of each treatment group. (**E**) Blood glucose concentration during intraperitoneal glucose tolerance test (ipGTT) (2 g/kg) of each treatment group. (**F**) Area under the curve (AUC) of blood glucose concentration during ipGTT of each treatment group. (**G**) Immunoreactive insulin (IRI) concentration during ipGTT (2 g/kg) of each treatment group. (**H**) AUC of IRI during ipGTT of each treatment group. * *p* < 0.05. Data are shown as mean ± SE.

**Figure 2 jcm-08-00674-f002:**
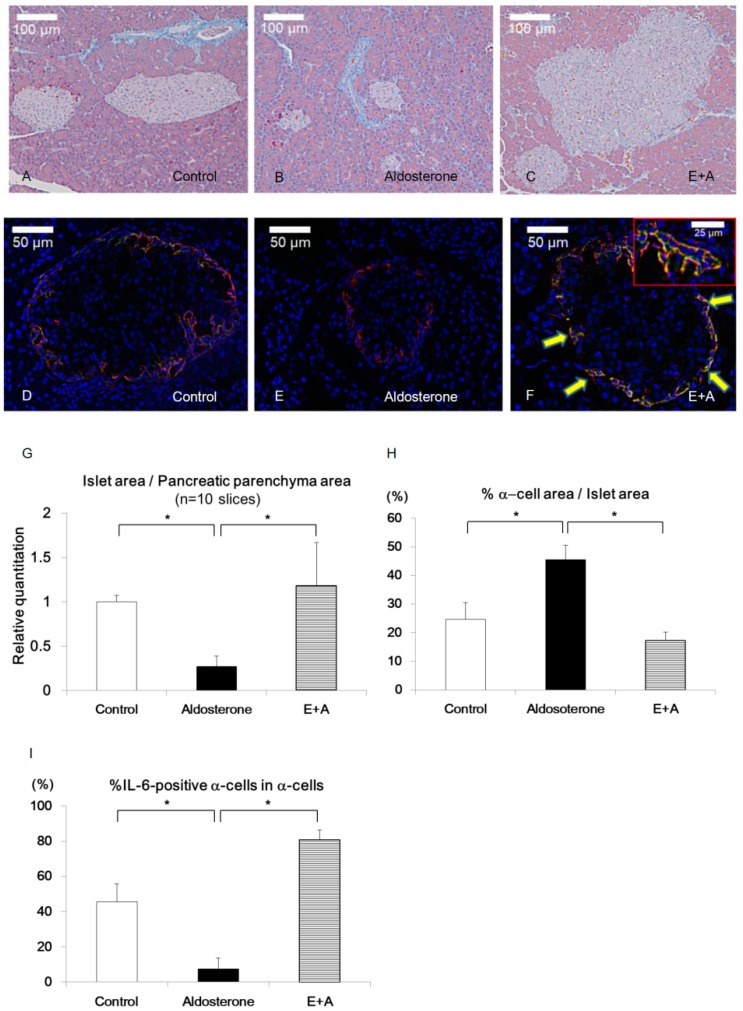
Histological analysis of pancreatic islets in PA model rodents. (**A**–**C**) Masson’s trichrome staining of the islets of each treatment group. (**D**–**F**) Double staining of glucagon (red) and IL-6 (green) of the islets of each treatment group. Arrows in F indicate IL-6-positive staining in α-cells. (**G**) Islet/pancreatic parenchyma area ratio of each group (*n* = 10 slices). (**H**) Percentages of α-cells area in islet area. (**I**) Percentages of IL-6-positive α-cells in α-cells. Scale bars in (**A**–**C**) = 100 μm and scale bars in (**D**–**F**) = 50 μm. * *p* < 0.05. Data are shown as mean ± SE.

**Figure 3 jcm-08-00674-f003:**
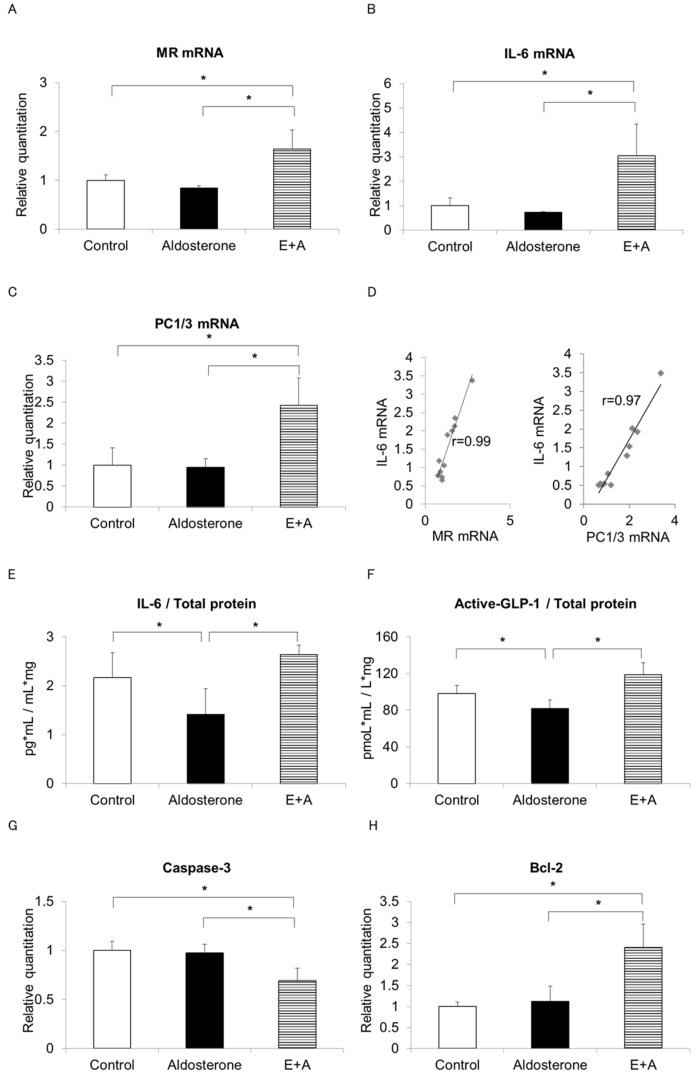
Alteration of mRNA levels and IL-6 and active GLP-1 concentrations in αTC cells. MR mRNA (**A**), IL-6 mRNA (**B**) and PC1/3 mRNA (**C**) were measured in αTC cells (p5), treated for 24 h with vehicle (Control), 10^−7^ M aldosterone (Aldosterone) or 10^−7^ M aldosterone after 1 h pre-treatment with 10^−5^ M eplerenone (E+A) (*n* = 4, each group). (**D**) Correlations between the relative quantity of MR mRNA and IL-6 mRNA (r = 0.99) or PC1/3 mRNA and IL-6 mRNA (r = 0.97) were plotted. IL-6 (**E**) and active GLP-1 (**F**) concentrations were determined by ELISA assays in supernatants from treatment groups. Relative mRNA levels of apoptosis-related caspase 3 (**G**) or anti-apoptosis Bcl2 (**H**) were measured in each group (*n* = 4, each). * *p* < 0.05. Data are shown as mean ± SE.

**Figure 4 jcm-08-00674-f004:**
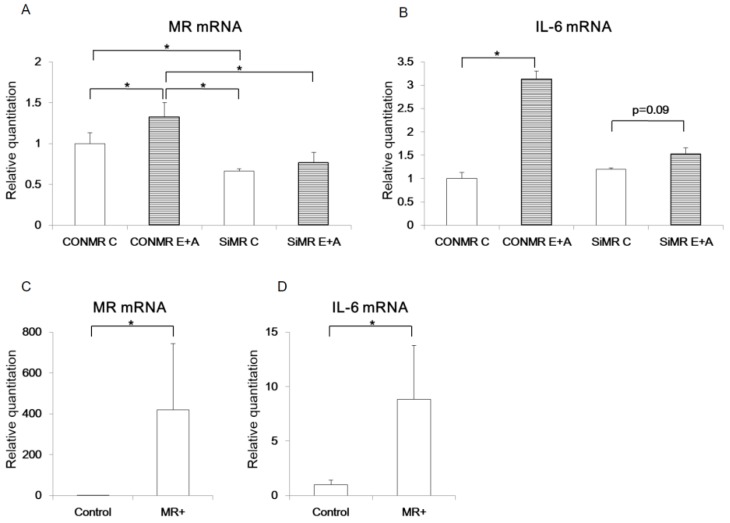
MR regulates IL-6 mRNA expression in pancreatic islet cells. MR mRNA (**A**) or IL-6 mRNA (**B**) were measured in αTC cells (p9) transfected with control siRNA (CONMR) or siRNA targeting MR (SiMR) and treated by vehicle (Control, C) or eplerenone and aldosterone (E+A) (*n* = 4). MR mRNA (**C**) and IL-6 mRNA (**D**) were measured in the 1C3 IKEI pancreatic cells (IKEI cells) (*p* = 3) transfected with empty vector (Control) or human MR-expressing vector (MR+) (*n* = 4). * *p* < 0.05. Data are shown as mean ± SE.

**Figure 5 jcm-08-00674-f005:**
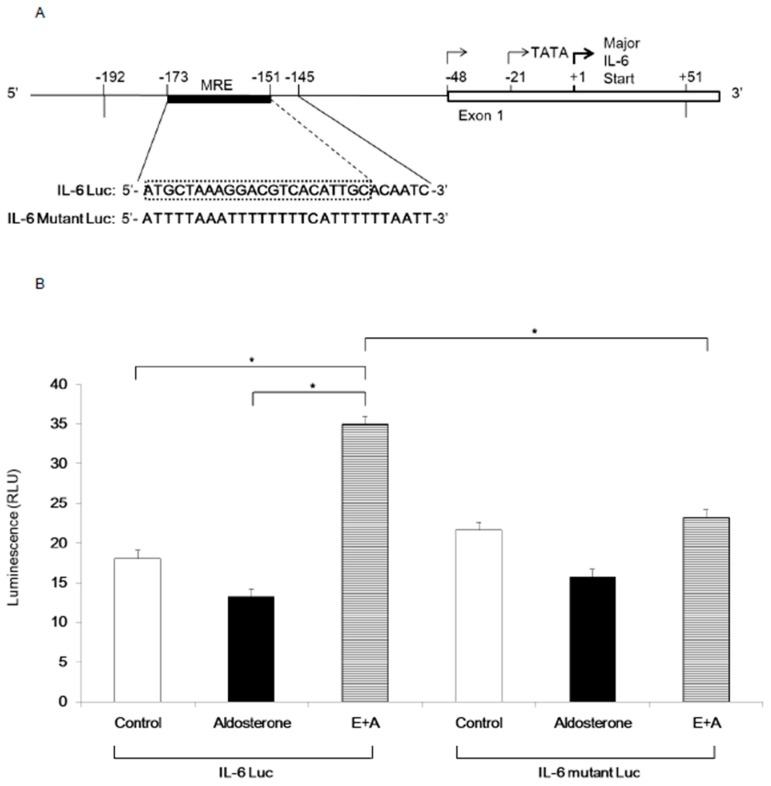
IL-6 expression in αTC cells upon eplerenone treatment requires MR binding element in the IL-6 gene promoter. (**A**) The map of the IL-6 gene 5’ flanking region is shown on top and the sequences of MRE are shown below (IL-6 Luc). The sequences of mutated MRE used for the experiment are shown below the IL-6 Luc (IL-6 Mutant Luc). The IL-6 promoter DNA fragments (−192 to +51) were ligated into the PGL-3 vector. Black box: MRE, the DNA sequences mutated in the IL-6 Mutant Luc are shown in red characters. White box: exon 1 of IL-6 gene. (**B**) Luciferase assays in αTC cells (p8) transfected with IL-6 Luc or IL-6 Mutant Luc and treated with control, aldosterone, E+A. * *p* < 0.05. (compared with each group of IL-6 Luc). Data are shown as mean ± SE.

**Figure 6 jcm-08-00674-f006:**
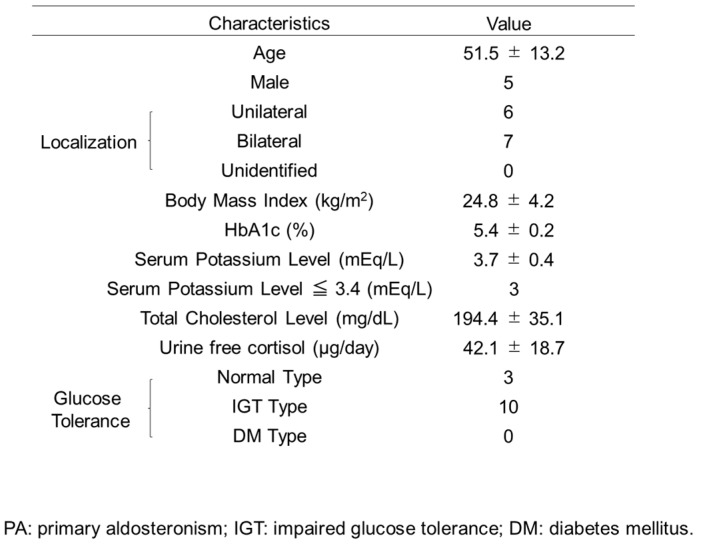
Baseline Characteristics of PA Patients (*n* = 13).

**Figure 7 jcm-08-00674-f007:**
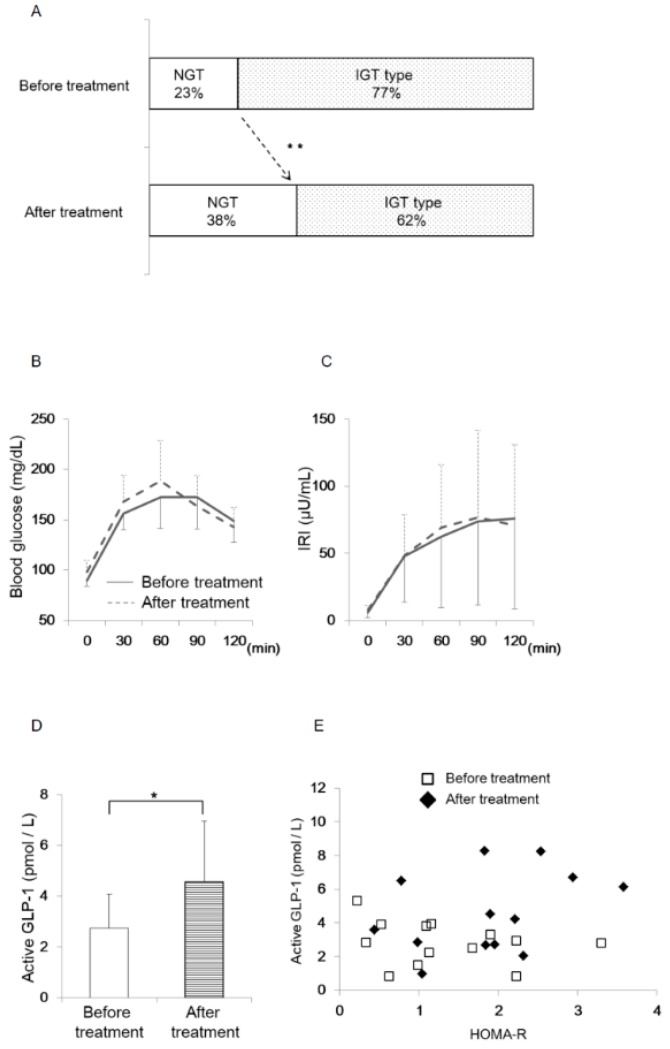
Glucose homeostasis in PA patients before and after eplerenone treatment (*n* = 13). (**A**) Results of oral glucose tolerance test (OGTT) before and after eplerenone treatment in PA patients (Chi-square test, ** *p* < 0.01). NGT: normal glucose tolerance. IGT: impaired glucose tolerance. (**B**) Blood glucose concentration before and after eplerenone treatment during OGTT. (**C**) Immunoreactive insulin (IRI) concentration before and after eplerenone treatment during OGTT. (**D**) Active GLP-1 concentration at 60 min during OGTT before and after eplerenone treatment (paired t-test, * *p* < 0.05). (**E**) Active GLP-1 concentration and HOMA-R per patient. Data are shown as mean ± SE.

**Figure 8 jcm-08-00674-f008:**
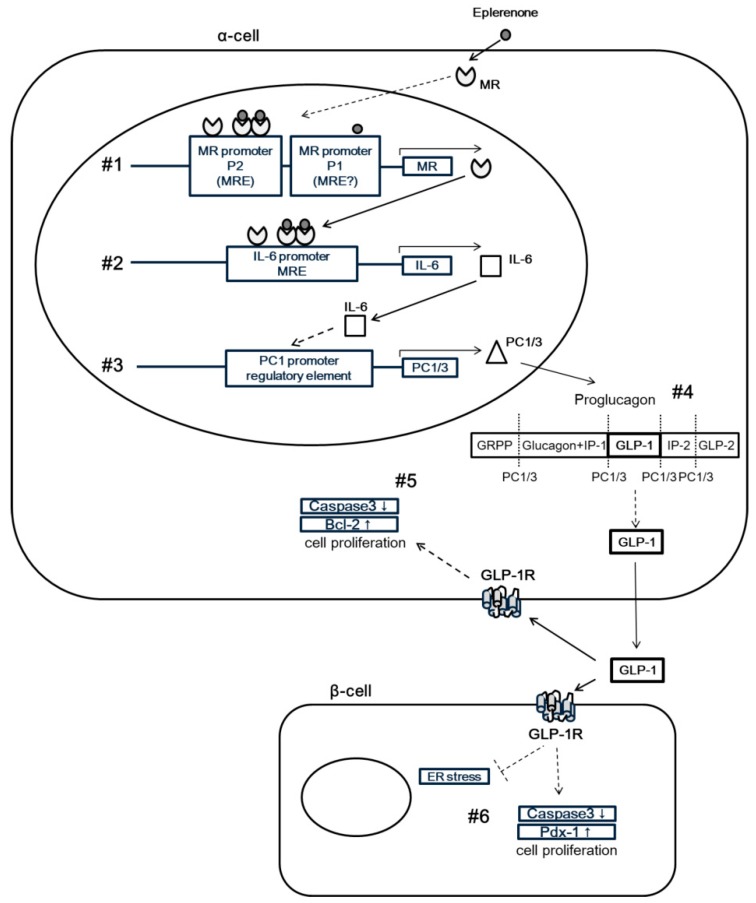
Proposed mechanism for MR in pancreatic islets. MR signal exerts a protective function in islets through IL-6 induction, which promotes active GLP1 secretion through PC1/3 activation. (1) MR induced from eplerenone, alone or in an eplerenone-MR complex, may bind to MRE in P2 in α-cells and promote MR expression. (2) MR and/or an eplerenone-MR complex may bind to the MRE in the IL-6 promoter to drive IL-6 expression. (3) IL-6 activation leads to increased PC1/3 mRNA levels and (4) induced PC1/3 may cleave proglucagon into GLP-1 in α-cells, resulting in elevated plasma levels of GLP-1. (5) Autocrine signaling of GLP-1 in α-cells may increase the expression of Bcl-2 mRNA and reduce the levels of caspase-3 mRNA, leading to protection of α-cells from apoptosis. (6) Paracrine signaling of GLP-1 in β-cells may increase the expression of Pdx-1 mRNA and reduce the levels of caspase-3 mRNA, thus protecting β-cells from endoplasmic reticulum stress and apoptosis.

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
