# Peer review of "Mineralocorticoid Receptor May Regulate Glucose Homeostasis through the Induction of Interleukin-6 and Glucagon-Like peptide-1 in Pancreatic Islets"

_jcm, 2019, doi:10.3390/jcm8050674_

Reviewer 1 Report

In this manuscript Goto and co-authors explore the role of MR and eplerenone on glucose homeostasis through an in vivo and in vitro study.

Some points need clarification

-          Please briefly detail how the diagnosis of primary aldosteronism was performed

-          Results, lines 200-201: the sentence should be rewritten. According to the presented data, IL-6 inhibition induced by aldosterone is reverted by the co-administration of eplerenone. Aldo + eplerenone do not induce IL-6 production, since according to Figure 1, panel I the group A+E is not significantly different from the control group.

-          Results 3.2 section. The authors state that eplerenone can induce IL-6 mRNA in vitro since its expression in the A+E group is increased over basal. However, to support this conclusion, the effects of eplerenone alone (without concomitant aldosterone treatment) on IL-6 mRNA expression should be included.

-          Results section 3.3 and Figure 4. Why the author decided to include only the control group and the E+A group? It should be extremely useful to see also the effects of aldosterone alone and eplerenone alone.

Author Response

-          Please briefly detail how the diagnosis of primary aldosteronism was performed

Line 97-101 According to the diagnosis of PA of the Japan Endocrine Society, the diagnosis of PA was performed using the result of plasma aldosterone concentration (pg/mL) / plasma renin activity (ng/mL/hr) ratio (ARR > 200), the captopril suppression test (ARR > 200 at 60 min after loading 50mg of captopril), and the adrenal vein sampling. 

-          Results, lines 200-201: the sentence should be rewritten. According to the presented data, IL-6 inhibition induced by aldosterone is reverted by the co-administration of eplerenone. Aldo + eplerenone do not induce IL-6 production, since according to Figure 1, panel I the group A+E is not significantly different from the control group.

Line 204-205 We rewrote the sentence to “These data indicate that IL-6 expression in pancreatic a-cells was reduced by the treatment of aldosterone and the treatment of eplerenone restored the IL-6 expression”.

-          Results 3.2 section. The authors state that eplerenone can induce IL-6 mRNA in vitro since its expression in the A+E group is increased over basal. However, to support this conclusion, the effects of eplerenone alone (without concomitant aldosterone treatment) on IL-6 mRNA expression should be included.

Thank you for your valuable comment. We have performed the requested experiment and rewrote the manuscript as follows. Line 233-236To investigate the effects of eplerenone alone, aTC cells (p8) were stimulated with vehicle (Control), 10-5 M eplerenone alone (Eplerenone) or 10-7 M aldosterone after pretreatment with 10-5 M eplerenone for 1 h (E+A). IL-6 mRNA was increased by 2.2-fold upon eplerenone treatment alone compared with the control, and PC1/3 mRNA showed a similar induction pattern as IL-6 (Supplementary Fig. 2B and C).

- Results section 3.3 and Figure 4. Why the author decided to include only the control group and the E+A group? It should be extremely useful to see also the effects of aldosterone alone and eplerenone alone.

→Although this suggestion is important, we did not investigate yet in this manuscript. Other experiments are currently planned. It should need some more time to complete. We decided to include only the control group and the E+A group to investigate whether the MR expression level alter the IL-6 expression level.

Reviewer 2 Report

The manuscript is well organized and the data are clearly presented and worthy of interest. The discussion is supported by the analysis of the results obtained. Comments: 1. To investigate the effect of MR blockade on IL-6 and GLP-1 production in pancreatic islets, the Authors performed an interesting in vitro experiment using alpha cells. In particular, they treated alphaTC cells with two different aldosterone concentrations (in order to avoid GR mediated confounding effect), as well as with a combination of aldosterone plus eplerenone. They observed that aldosterone  did not modulate IL-6 mRNA expression, despite its efficacy demonstrated by its effect on SGK-1 mRNA expression.  Whereas aldosterone plus eplerenone induced a significant increase in IL-6 gene expression (supplementary fig.2B). The same pattern is observed in apoptosis gene analysis. It would be of interest to evaluate the effect of eplerenone alone, given the absence of aldosterone effect. Is this a ligand-independent mechanism? The Authors should better discuss the discrepancy between gene expression analysis and the reduction observed in IL-6 and GLP-1 release in the supernatants. 2. In conclusions, the Authors should better discuss the potential therapeutic effect of eplerenone as MR antagonist in diabetes.

Author Response

Comments: 1.To investigate the effect of MR blockade on IL-6 and GLP-1 production in pancreatic islets, the Authors performed an interesting in vitro experiment using alpha cells. In particular, they treated alphaTC cells with two different aldosterone concentrations (in order to avoid GR mediated confounding effect), as well as with a combination of aldosterone plus eplerenone. They observed that aldosterone did not modulate IL-6 mRNA expression, despite its efficacy demonstrated by its effect on SGK-1 mRNA expression. Whereas aldosterone plus eplerenone induced a significant increase in IL-6 gene expression (supplementary fig.2B). The same pattern is observed in apoptosis gene analysis. It would be of interest to evaluate the effect of eplerenone alone, given the absence of aldosterone effect. Is this a ligand-independent mechanism? The Authors should better discuss the discrepancy between gene expression analysis and the reduction observed in IL-6 and GLP-1 release in the supernatants.

Line233-236) To investigate the effects of eplerenone alone, aTC cells (p8) were stimulated with vehicle (Control), 10-5 M eplerenone alone (Eplerenone) or 10-7 M aldosterone after pretreatment with 10-5 M eplerenone for 1 h (E+A). IL-6 mRNA was increased by 2.2-fold upon eplerenone treatment alone compared with the control, and PC1/3 mRNA showed a similar induction pattern as IL-6 (Supplementary Fig. 2B and C). These results suggest that the eplerenone-MR complex or MR alone (ligand-independent mechanism) may drive IL-6 expression through the MRE.  

Line391-402) The proinflammatory cytokines which lacks a signal peptide are not fully understood the mechanisms of  those secretion. Previous report revealed that microvesicle shedding is a major secretory pathway for rapid IL-1b release from activated monocytes, and may represent more general mechanism for secretion of similar leaderless secretory proteins [1]. In our study, IL-10 mRNA expression was significantly increased in the aldosterone group compared with the control and the E+A group in aTC cells. IL-10 altered the number of autophagosome (microvesicle) in myocytes treated with Angiotensin II [2]. In addition, previous reports suggest that activation of IL-10 receptors leads to the inhibition of IL-6 release [3] [4]. Thus IL-6 and GLP-1 release in the supernatants may be decreased by IL-10 mediated mechanism in the aldosterone group. More detailed mechanisms should be clarified in future. 

2. In conclusions, the Authors should better discuss the potential therapeutic effect of eplerenone as MR antagonist in diabetes.

We added the sentence in conclusions. Line508-509) Eplerenone may have the potential benefit for diabetic therapy with its effect of increasing GLP-1 secretion in a-cells.

1.         MacKenzie, A.; Wilson, H.L.; Kiss-Toth, E.; Dower, S.K.; North, R.A.; Surprenant, A. Rapid secretion of interleukin-1beta by microvesicle shedding. Immunity 2001, 15, 825-835.

2.         Samanta, A.; Dawn, B. Il-10 for cardiac autophagy modulation: New direction in the pursuit of perfection. Journal of molecular and cellular cardiology 2016, 91, 204-206.

3.         Sabat, R.; Grutz, G.; Warszawska, K.; Kirsch, S.; Witte, E.; Wolk, K.; Geginat, J. Biology of interleukin-10. Cytokine & growth factor reviews 2010, 21, 331-344.

4.         Hempel, L.; Korholz, D.; Bonig, H.; Schneider, M.; Klein-Vehne, A.; Packeisen, J.; Mauz-Korholz, C.; Burdach, S. Interleukin-10 directly inhibits the interleukin-6 production in t-cells. Scandinavian journal of immunology 1995, 41, 462-466.

Round  2

Reviewer 1 Report

The reviewer commemnts have been addressed

Author Response

We appreciate your kind comments to improve our manuscript.

Reviewer 2 Report

Lines 391-393

The following sentences don't make sense. Please change "The discrepancy between gene expression analysis and the reduction is observed in IL-6 and GLP-1 release in the supernatants. The proinflammatory  cytokines which lacks a signal peptide are not fully understood the mechanisms of those secretions." with " The discrepancy observed between increased IL-6 and GLP-1 gene expression and reduced release in supernatants may be due to the fact that the mechanisms regulating the pronflammatory cytokines, without signal peptide, are not fully understood".

Author Response

Lines 391-393

The following sentences don't make sense. Please change "The discrepancy between gene expression analysis and the reduction is observed in IL-6 and GLP-1 release in the supernatants. The proinflammatory  cytokines which lacks a signal peptide are not fully understood the mechanisms of those secretions." with " The discrepancy observed between increased IL-6 and GLP-1 gene expression and reduced release in supernatants may be due to the fact that the mechanisms regulating the pronflammatory cytokines, without signal peptide, are not fully understood".

→ Thank you for your suggestion. We changed the sentences according to your suggestion as line 398-400.
